# Insights into the Possible Molecular Mechanisms of Resistance to PARP Inhibitors

**DOI:** 10.3390/cancers14112804

**Published:** 2022-06-05

**Authors:** Claudia Piombino, Laura Cortesi

**Affiliations:** Genetic Oncology Unit, Department of Oncology and Haematology, University Hospital of Modena, 41125 Modena, Italy; 256171@studenti.unimore.it

**Keywords:** PARP inhibitors, BRCA1, BRCA2, homologous recombination, non-homologous end joining, fork stabilization

## Abstract

**Simple Summary:**

The increasingly wide use of PARP inhibitors in breast, ovarian, pancreatic, and prostate cancers harbouring a pathogenic variant in *BRCA1* or *BRCA2* has highlighted the problem of resistance to therapy. This review summarises the complex interactions between PARP1, cell cycle regulation, response to stress replication, homologous recombination, and other DNA damage repair pathways in the setting of *BRCA1/2* mutated cancers that could explain the development of primary or secondary resistance to PARP inhibitors.

**Abstract:**

PARP1 enzyme plays an important role in DNA damage recognition and signalling. PARP inhibitors are approved in breast, ovarian, pancreatic, and prostate cancers harbouring a pathogenic variant in *BRCA1* or *BRCA2*, where PARP1 inhibition results mainly in synthetic lethality in cells with impaired homologous recombination. However, the increasingly wide use of PARP inhibitors in clinical practice has highlighted the problem of resistance to therapy. Several different mechanisms of resistance have been proposed, although only the acquisition of secondary mutations in *BRCA1/2* has been clinically proved. The aim of this review is to outline the key molecular findings that could explain the development of primary or secondary resistance to PARP inhibitors, analysing the complex interactions between PARP1, cell cycle regulation, PI3K/AKT signalling, response to stress replication, homologous recombination, and other DNA damage repair pathways in the setting of *BRCA1/2* mutated cancers.

## 1. Introduction

Poly-(ADP-ribose) polymerase (PARP) enzyme PARP1 plays an important role in DNA damage recognition and signalling, as it binds single-stranded DNA breaks (SSBs) and then organizes their repair by synthesising PAR chains on target proteins (the so-called PARylation) [1]. In detail PARP1, once bound to SSBs via N-terminal zinc-finger DNA-binding domain, catalyses the polymerization of ADP-ribose moieties from NAD+ on target proteins, mainly PARP1 itself and histones. This process leads to chromatin relaxation and recruitment of other DNA repair enzymes such as XRCC1 [2,3,4,5]. The scaffolding protein XRCC1 stimulates the DNA kinase and DNA phosphatase activities of polynucleotide kinase at SSBs, accelerating the base excision repair reaction [6]. It is also reported that PARP1 promotes recruiting of MRE11, ATM and BRCA1, which are involved in double-stranded DNA break (DSB) repair by homologous recombination (HR) [7,8,9]. While PARP1 DNA binding is independent of its catalytic activity, dissociation of PARP1 from DNA requires PARylation presumably through a steric mechanism due to highly negatively charged PAR chains [10].

PARP inhibitors (PARPi) act mainly in a double way. The first proposed mechanism is the inhibition of the catalytic activity of PARP1, which results in synthetic lethality in cells with impaired HR [11,12,13]. In fact, inhibition of PARP1 promotes SSBs, which, if unrepaired, consequently lead to DSBs by collapse of the stalled replication fork during DNA replication [14]. In eukaryotic cells, DSBs are mainly resolved by the error-free mechanism of HR, which uses the intact sister chromatid as a template. HR deficiency induces activation of the more error-prone template-independent non-homologous end-joining (NHEJ) pathway, therefore, together with PARPi causing genomic instability followed by cell death [15,16]. Subsequent studies have revealed that most PARPi cause cytotoxicity by trapping PARP1 at sites of DNA damage [17,18,19]. According to the hypothesis proposed by Murai et al. [18], PARPi binding to the catalytic domain of PARP1 allosterically modifies interactions between DNA and the N-terminal DNA-binding domain of the protein, to the point that PARP1 becomes trapped on DNA. More recently, a third mechanism of PARPi sensitivity has been identified [20,21]. In cells with HR deficiency, aside from the NHEJ pathway, DSBs can be repaired by the microhomology-mediated end joining (MMEJ or Alt-EJ) pathway. Similarly to NHEJ, MMEJ is intrinsically error-prone, as the use of regions of microhomology inside DNA leads to deletions of nucleotides from the strand being repaired and to chromosomal translocations. In this pathway, the efficient recruitment of the DNA polymerase POLQ to the DSB requires PARP1. A PARPi will, therefore, block the MMEJ pathway and cause HR-deficient cell death.

PARP inhibitors are actually approved in breast, ovarian, pancreatic and prostate cancers harbouring a pathogenic or likely pathogenic variant in *BRCA1* or *BRCA2* (*BRCA1/2*) [22,23,24]. *BRCA1/2* mutation prevalence varied widely from 1.8% in sporadic breast cancer to 36.9% in estrogen receptor/progesterone receptor low HER2 negative breast cancer [25]. Germline mutations in *BRCA1*/*2* have been identified in 13–15% of women diagnosed with ovarian cancer, and somatic mutations are found in an additional 7% [26,27,28]. Germline *BRCA1/2* mutations can be found in up to 8% of patients with sporadic pancreatic cancer [29]. In a sample of 692 patients with metastatic prostate cancer, unselected for family history or age at diagnosis, 5.3% carried a *BRCA2* mutation, and 0.9% carried a *BRCA1* mutation [30]. The increasingly wide use of PARPi in clinical practice is highlighting the problem of resistance to therapy. Considering the complex interaction between PARP1, HR and other DNA damage repair pathways in the setting of *BRCA1/2* mutated cancers, several different mechanisms of resistance have been proposed, although some of them have been only described preclinically. The aim of this review is to outline the key molecular findings that could explain the mechanisms of primary or secondary resistance to PARPi (summarised in Table 1).

## 2. Primary Resistance

Primary resistance can be defined by a complete lack of response to the therapy. It can be mediated by co-occurring mutations in genes other than *BRCA1/2* involved in cell cycle regulation and DNA damage repair, which could overcome HR deficiency, thus making cancer cells resistant to PARPi. Several mechanisms have been suggested but only in the preclinical setting.

### 2.1. PTEN Deficiency and PI3K/AKT Pathway Activation

Phosphatase and tensin homolog (PTEN) is a dual protein/lipid phosphatase that acts as a tumour suppressor gene inhibiting the PI3K/AKT pathway. In detail, PTEN converts phosphatidylinositol 3,4,5-triphosphate (PIP3) into phosphatidylinositol 4,5-bisphosphate (PIP2), antagonising PI3K action and preventing AKT activation and consequent cell growth and cell proliferation [62,63]. *PTEN* is often inactivated in different cancers [64]. Although AKT activation promotes BRCA1 expression through phosphorylation [65], it has been shown that BRCA1 can downregulate AKT activation in different ways: by acting on upstream kinases of AKT [31,66], by ubiquitin-mediated proteasomal degradation or by activating a protein serine/threonine phosphatase PP2A in breast cancer cells (Figure 1) [67]. Therefore, negative mutations and/or reduced expression of *BRCA1* may activate the PI3K/AKT pathway [67]. Significantly, PTEN loss is highly associated with *BRCA1*-defective breast cancers, probably due to genomic instability resulting from deficient DSB repair [68], and the resulting PI3K/AKT activation stimulates the growth of those cancers [69].

Apart from its role in regulating the PI3K/AKT pathway, PTEN loss of function causes genetic instability, as *PTEN-null* cells lack competent HR DNA repair. In detail, PTEN has been suggested to contribute to RAD51 expression [70,71,72]. Dedes et al. [32] demonstrated that endometrioid endometrial cancer cell lines lacking PTEN function are unable to elicit competent HR DNA repair and are relatively sensitive to PARPi, and, as a consequence, *PTEN* silencing significantly increases the sensitivity to PARPi reducing RAD51 foci formation. Re-expression of *PTEN* in PTEN-mutant endometrioid endometrial cancer cells leads to relative resistance to PARPi [32].

Together, these observations corroborate the hypothesis that PTEN loss, which is highly associated with *BRCA1*-defective breast cancer, contributes to PARPi sensitivity. On the other hand, wild-type *PTEN* tumours could demonstrate relative resistance to PARPi. Considering that the PI3K/AKT pathway is constitutively active in *BRCA1*-defective human cancer cells [73], the combination of PTEN-related PI3K/AKT pathway inhibitors such as perifosine with DNA topoisomerase I (TOP1) or PARPi results in enhanced cancer cell killing in these tumours [74,75], suggesting a new possibly targetable pathway in case of PARPi resistance. A phase I trial of the PARPi Olaparib with the AKT inhibitor capivasertib, involving patients with advanced solid tumours carrying a germline *BRCA1/2* mutation or somatic DNA damage response or PI3K/AKT pathway alterations, demonstrated the safety, tolerability and pharmacokinetic-pharmacodynamic activity of this combination [76].

### 2.2. ATM Roles and Loss of NHEJ Pathway

The choice between NHEJ and HR to repair DSBs is determined by several mechanisms, including activation of HR by cyclin-dependent kinase (CDK) activity (while NHEJ operates throughout interphase, HR is restricted to the S and G2 phase of the cell cycle, when a homologous sister chromatid is available as template) [16], or direct competition between HR and NHEJ stimulating factors at DSB sites [77]. During G2/S, HR is activated by binding of the MRE11–RAD50–NBS1 (MRN) complex to DSB ends. The MRN complex initiates DNA end resection, leading to the formation of single-strand DNA (ssDNA) at the extremity of the DSB. The ssDNA is protected from degradation by the loading of replication protein A (RPA) [78]. CtIP phosphorylated by CDK binds MRN complex to facilitate end resection [79]. The MRN complex recruits and activates the protein kinase ATM [80], while the sensor protein RPA finally drives ATR activation [81]. ATM and ATR phosphorylate several proteins involved in the HR [82]. The tumour suppressor complex BRCA1-BARD1 phosphorylated by ATM facilitates DNA end resection and interacts with PALB2, which in turn promotes the recruitment of BRCA2 [54]. PALB2 and BRCA2 remove RPA and facilitate the assembly of the RAD51 recombinase nucleoprotein filament. RAD51 nucleoprotein filament mediates the invasion of ssDNA into the intact sister chromatid, searching for a homologous template for DNA synthesis and repair [83,84,85]. During G1/2, 5′-3′ end resection is suppressed, and HR is inhibited due to the lack of sister chromatid [86]. ATM phosphorylates 53BP1 in multiple residues [87]. Once phosphorylated, 53BP1 binds and recruits RIF1 and PTIP that, together with the downstream effectors REV7 and Sheldin, inhibit 5′-3′ end resection and promote NHEJ [88]. Classical NHEJ starts when Ku70/80 heterodimer binds DSBs and recruits the DNA-dependent protein kinase catalytic subunit (DNA-PKcs) to form the DNA–PK complex. The latter engages XRCC4 and DNA ligase IV (LIG4) to align and ligates DNA ends regardless of sequence homology [89]. BRCA1 antagonizes 53BP1 by limiting its interaction with the chromatin in the S phase and stopping the translocation of RIF1 to DSBs in the G2/S phase, promoting HR [90,91]. Additionally, the different recruitment kinetics of the MRN and Ku complexes, which activate HR and NHEJ repair, respectively [92], as well as MRN/CtIP-dependent removal of Ku complex from DSBs [93], influence the choice between the two pathways (Figure 2).

Balmus et al. [33] demonstrated that ATM orchestrates the response to DSBs, promoting HR and preventing NHEJ that would result in chromosomal aberrations. They performed genome-wide CRISPR-Cas9 screens in wild-type and *Atm-null* mouse embryonic stem cells treated with the TOP1 inhibitor topotecan. They established that, in the absence of ATM, NHEJ-mediated repair of DSBs induced by TOP1 or PARPi leads to aberrant chromatid fusions and cell death. This could be avoided by impairing NHEJ through inactivation of the terminal components of the NHEJ machinery or promoting HR by inactivation of the BRCA1-A complex (BRCC45, ABRAXAS, MERIT40, RAP80, and BRCC36). In detail, the toxicity is mainly determined by the LIG4-mediated ligation step of NHEJ at DSBs and not by defective Ku removal, as previously suggested [93]. ATM phosphorylates hundreds of substrates [82], so it may operate at multiple levels to prevent NHEJ and ensure HR. The authors finally suggested that loss or reduced expression of the genes for the BRCA1-A complex and of the NHEJ components Ku70/80, XRCC4, XLF, and LIG4 in *ATM*-deficient tumours could explain resistance to PARPi in some cases [33].

The need for competent NHEJ to ensure synthetic lethality by PARPi in HR defective cells has been also proved by other studies. Patel et al. [34] disabled NHEJ by using genetic or pharmacologic approaches and rescued the lethality of PARP inhibition or down-regulation in cell lines lacking BRCA2, BRCA1, or ATM. McCormick and colleagues [35] assessed NHEJ in a panel of ovarian cancer cell lines and primary ascitic-derived ovarian cancer cultures, finding that NHEJ-defective cultures were resistant to the PARPi Rucaparib and only NHEJ-competent/HR-defective cultures were sensitive.

### 2.3. ALC1 Overexpression

Amplified in liver cancer 1 (ALC1) is a PAR-dependent chromatin remodeler that directly binds PAR chains, promoting PARP1 activation [94,95]. In a genome-wide CRISPR knockout screen with Olaparib, Juhász and colleagues [36] identified ALC1 as a key modulator of sensitivity to the PARPi Olaparib. ALC1 deficiency stimulates trapping of inhibited PARP1, which then impairs the binding of both the HR and NHEJ repair factors to DNA lesions. Moreover, they established that ALC1 overexpression, which is a common trait of many solid cancers, often associated with poor prognosis [96], reduces the sensitivity of *BRCA1/2*-deficient cells to PARPi. Analysis of the ALC1 expression levels before the use of PARPi could predict a mechanism of primary resistance to the therapy.

## 3. Secondary Resistance

Secondary or acquired resistance can be defined as the onset of the lack of response to treatment despite being successfully used before. Under therapeutic selective pressure, resistance to cytotoxic or target agents can emerge due to the expansion of pre-existing subclonal populations or from the evolution of drug-tolerant cells [97]. Several mechanisms of acquired resistance to PARPi have been suggested, although only reversion mutations in *BRCA1/2* have been proved in the clinical setting.

### 3.1. Upregulation of ABC Transporters

Increased expression of the membrane-bound ATP-dependent efflux pump P-glicoprotein, encoded by *ABCB1* (MDR1), is one of the most well-characterised mechanisms of multidrug resistance [98], in particular for doxorubicin, paclitaxel and related taxane drugs [99,100,101]. A chromosomal amplification event at 7q11.2-21 has been correlated to increased *ABCB1* copy number and consequent P-glycoprotein expression in paclitaxel-resistant cancer cells [102].

Using a mouse model of *BRCA2*-deficient sarcomatoid breast cancer, Jaspers et al. [37] found that multidrug resistance was strongly associated with high expression of the ABCB1, known to be efflux transporter of Olaparib [103], docetaxel [104], and doxorubicin [105]. In novel A2780-derived ovarian cancer cell lines, paclitaxel-resistant cells were cross-resistant to the PARPi Olaparib and Rucaparib but not to Veliparib. This resistance correlated with increased ABCB1 expression and was reversible following treatment with ABCB1 inhibitors [38]. These findings, although cell-line based, could help in PARPi choice in paclitaxel-resistant patients, especially in second-line or maintenance therapy.

### 3.2. Decreased PARP1 Trapping

Through a genome-wide and high-density CRISPR-Cas9 mutagenesis screen to identify mouse embryonic stem cell mutants resistant to the PARPi Talazoparib, Pettitt and colleagues [39] identified point mutations in the N-terminal zinc-finger domain of PARP1 that, abolishing DNA binding, cause PARP1 resistance and alter PARP1 trapping. In support of this finding, they also observed a PARP1 mutation that abolished trapping in a patient with de novo resistance to Olaparib. In this experimental model, also mutations in amino-acid residues involved in hydrogen-bonding interactions that bridge the DNA-binding domain and the catalytic domain can cause PARPi resistance, likely by impairing PARP1 trapping; this reinforces the observations that inter-domain interactions are critical for PARP1 binding and activation [2,3,4].

PAR glycohydrolase (PARG) is the main enzyme responsible for degrading PAR chains, counteracting PARylation catalysed by PARP1 [106]. By combining genetic screens with multi-omics analysis of matched PARPi sensitive and resistant *BRCA2*-mutated mouse mammary tumours, Gogola et al. [40] identified loss of PARG as a major resistance mechanism. In a panel of 34 PARPi resistant tumours, they observed decreased expression of *Parg* in 17 tumours and acquired copy-number loss of the *Parg* locus in 22 tumours. Moreover, they demonstrated that PARG depletion does not increase PARP1 trapping per se but prevents excessive PARP1 binding to chromatin, thus reducing PARPi-dependent accumulation of toxic PARP1–DNA complexes. As a confirmation of this, PARG depletion also resulted in resistance to Talazoparib, a highly potent PARP1–DNA trapping agent. Finally, they found PARG-negative cell clones in a subset of human serous ovarian and triple-negative breast cancers (TNBCs), suggesting that this mechanism can contribute to PARPi resistance in vivo.

### 3.3. Restoration of HR

#### 3.3.1. Reversion Mutations in BRCA1/2

The only mechanism of secondary resistance to PARPi currently validated in clinical practice is the acquisition of secondary mutations in *BRCA1/2* restoring the open reading frame and detected upon treatment progression (Figure 3). Tobalina and colleagues [41] analysed sequencing data available in the literature from tumour or circulating tumour DNA (ctDNA) of 327 patients with tumours harbouring mutations in *BRCA1/2* (234 patients with ovarian cancer, 27 with breast cancer, 13 with pancreatic cancer, 11 with prostate cancer and 42 with a cancer of unknown origin) on progression after platinum or PARPi treatment. Secondary mutations leading to the correction of the original mutation or to restoration of the *BRCA1/2* open reading frame and functional proteins that were disrupted by it, the so-called reversion mutations, have been identified in 86 patients (26%). A comprehensive analysis of the reversion mutations points out that most amino acid sequences encoded by exon 11 in *BRCA1* and *BRCA2* are essential with regard to the protein function and consequentially to generate resistance to platinum or PARPi. Deletions accounted for the majority of secondary mutations, probably due to the upregulated use of the error-prone mechanisms of NHEJ or MMEJ to repair DNA DSBs in HR-deficient cells. As proof of this, researchers observed an increase in the length of micro-homologies (that can be considered MMEJ signatures) surrounding breakpoints in secondary deletion when compared with primary mutation and, in the case of *BRCA2* deletions (regardless of being primary or secondary), compared with those in *BRCA1*, suggesting that several variables such as chromosomal location and/or chromatin landscape around *BRCA1* and *BRCA2* could determine the choice of the repair pathway when these genes are altered [107].

#### 3.3.2. Hypomorphic BRCA1 Allele

It is unclear whether all pathogenic *BRCA1* mutations have comparable effects on the response to therapy. C-terminal truncated BRCA1 proteins can be semifunctional, retaining the protein domains necessary to mediate interactions with PALB2-BRCA2 (Figure 3). Under PARPi selection pressure, the HSP90 protein can interact with and stabilize these mutant BRCA1 proteins, and the partial BRCA1 function can promote RAD51 loading onto DNA following DNA damage [108,109,110,111,112,113]. Even some N-terminal *BRCA1* mutations may retain some residual activity in DNA damage response (Figure 3). Comparing two different *BRCA1* mutations on N-terminal domain (185delAG and 5382insC) in genetically engineered mouse models, *BRCA1* (185delAG) tumours responded significantly worse to PARPi than the *BRCA1* (5382insC) tumours. The hypothesis suggested is that *BRCA1* (185delAG) tumour cells produce a RING-less BRCA1 protein; this structure may lead to PARPi resistance through its residual activity by activating RAD51 [42].

Additionally, *BRCA1* mRNA isoforms originated by alternative splicing and lacking exon 11 can produce truncated but hypomorphic proteins that have residual BRCA1 function (Figure 3). Ovarian cancer cells with BRCA1 splice isoforms lacking exon 11 may be clonally selected under PARPi treatment. Analysis of clinical ovarian cancer samples showed that exon 11 mutation carriers had worse overall survival when compared with non-exon 11 mutation carriers, probably due to less sensitivity to platinum-based chemotherapy by the cells with residual BRCA1 function of the hypomorphic protein [43]. The ability of BRCA1 hypomorphic protein to confer resistance to drug treatment has also been described in patient-derived xenograft (PDX) models [44,45,46].

#### 3.3.3. Epigenetic Reversion of BRCA1 Promoter Hypermethylation

Epigenetic inactivation of *BRCA1* through promoter hypermethylation occurs in approximately 15% of TNBCs and 11% to 15% of serous ovarian cancers [114,115]. In PDX models of *BRCA1*-deficient breast cancer, responses of the *BRCA1*-methylated PDX tumours to alkylating agents and Olaparib were comparable with those of the *BRCA1*-mutated model. However, several *BRCA1*-methylated PDX tumours acquired therapy resistance via re-expression of *BRCA1* thanks to the loss of the *BRCA1* promoter methylation [47]. This is consistent with previous cell line studies [48,49] and to what was shown in a patient with ovarian carcinoma [115]. Interestingly, in other *BRCA1*-methylated PDX tumours, *BRCA1* re-expression was determined by de novo intrachromosomal genomic rearrangements through which *BRCA1* transcription is placed under the control of a heterologous promoter [47].

#### 3.3.4. Loss of End Resection Regulation

The tumour suppressor 53BP1 plays an important role in DSB repair pathway choice [82,83,116]. RIF1 is the most proximal effector of 53BP1 in its anti-resection function of DNA 5′ ends [117]; REV7 is a downstream factor in 53BP1-/RIF1-dependent NHEJ promotion [118]. Genetic depletion of *53BP1*, *RIF1*, or *REV7* provides synthetic viability to *BRCA1*-deficient cells and provides resistance to PARPi, restoring HR. Belotserkovskaya et al. [50] demonstrated that depleting 53BP1 in *BRCA1-null* cell lines reversed synthetic lethality mediated by PARPi, reactivating HR in a PALB2-dependent manner. Loss of REV7 re-establishes CtIP-dependent end resection of DSBs in *BRCA1*-deficient cells, leading to HR restoration and PARPi resistance [51]. Additionally, inactivating mutations in Sheldin, a 53BP1 effector complex, causes resistance to PARPi in *BRCA1*-deficient cells and cancers due to restoration of HR [52,53]. Interestingly, in all of these experiments, inhibition of NHEJ selectively reverts the *BRCA1*-depleted cell lines but not the *BRCA2* mutant lines, reflecting the different roles of BRCA1 and BRCA2 in the DNA-damage repair pathways. As mentioned before, BRCA1 antagonizes 53BP1 and RIF1, avoiding NHEJ, so HR can still function in the absence of BRCA1, but cells will mainly proceed down the NHEJ pathway. Conversely, BRCA2 and PALB2 roles are within the HR pathway itself, so even inhibiting NHEJ, HR cannot work, leading to cell death.

Dynein light chain 1 (DYNLL1), like 53BP1, inhibits DNA end resection, promoting NHEJ. Although the exact relationship between DYNLL1 and the 53BP1 axis is still unclear [119,120], loss of DYNLL1 was associated with PARPi resistance in a panel of patient-derived *BRCA1*-mutant high-grade serous ovarian cancer lines [54].

#### 3.3.5. Overexpression of RAD51 and RAD51 Paralogs

Sequencing core HR pathway genes in 12 pairs of pre-treatment and post-progression tumour biopsy samples collected from ovarian cancer patients treated with rucaparib, Kondrashova and colleagues [55] identified a truncation mutation in *BRCA1, RAD51C,* or *RAD51D* in 6 of 12 pre-treatment biopsies and one or more secondary mutations that restored the open reading frame in 5 of 6 paired post-progression biopsies; four distinct secondary mutations were observed for *RAD51C*. These mutations, as well as mutations in RAD51 itself identified in *BRCA1*-mutant TNBCs [56], are thought to restore RAD51 ability to mediate homologous sequence invasion during HR, cancelling the synthetic lethal effects of PARPi treatment.

Similarly, downregulation of EMI1 is associated with RAD51 stabilization. In detail, EMI1 assembles a ubiquitin ligase complex that constitutively targets RAD51 for degradation. A subset of *BRCA1*-mutant TNBC cells have proven to develop resistance to PARPi by downregulating EMI1 and restoring RAD51-dependent HR [57].

### 3.4. Stabilization of Stalled Forks

The HR pathway is not only involved in DNA DSB repair, but also in replication fork repair and restart [121]. During DNA duplication, replication forks often encounter obstacles that can lead to genotoxic fork stalling [122]. Cells have acquired several processes to maintain genome integrity by stabilizing, repairing and restarting stalled forks, thus preventing a possible mechanism of tumorigenesis [123,124]. Stalled forks are characterized by exposed DNA ends in the form of single- or double-strand DNA, which can be degraded by various cellular nucleases, including MRE11 and CtIP [125]. Under replication stress, extensive ssDNA is readily protected and stabilised by RPA; RPA phosphorylation by ATR and DNA-PKcs promotes PALB2 and BRCA2 translocation to stalled replication forks [126,127,128]. The RPA–ssDNA complex is then displaced by the RAD51 protein [129]. BRCA2 and BRCA1–BARD1 complex enhance the formation of stable RAD51 nucleoprotein filaments, which mediates replication fork reversal. The phosphorylation-directed prolyl isomerase PIN1 enhances BRCA1–BARD1 interaction with RAD51 (Figure 4). This mechanism is fundamental to regulate fork degradation mediated by the MRE11 nuclease [130,131,132,133,134]. In fact, in BRCA1/2-proficient cells, reversed forks are resected in a controlled manner, promoting HR-dependent fork repair and restart; in BRCA1/2-deficient cells, regressed forks are processed more extensively by MRE11 or converted into DSBs and can only be rescued by break-induce replication, a highly mutagenic process [121,125,132,135,136]. As proof that HR and replication fork protection are distinctive processes, the RAD51-binding region in the TR2 domain of BRCA2 or isomerization of the RING domain of BRCA1 are required for their function in DNA replication fork protection, but not in HR [130,134].

Like the BRCA2–RAD51 axis, also the Fanconi Anemia (FA) components FANCB and FANCD2 suppress MRE11-mediated fork degradation in a way linked to RAD51 protein [58,133,137]. FANCD2 is significantly overexpressed in *BRCA1/2*-mutated breast or ovarian cancers, highlighting its importance in limiting replication stress [58,138].

PARP1 activation at stalled replication forks contributes to MRE11 recruitment [136]; in addition, together with DNA-PKcs, PARP1 collaborates to engage XRCC1 for fork repair and restart, which suggests an involvement of the NHEJ machinery in stalled fork protection [139]. Since MRE11 recruitment by PARP1 is responsible for extensive fork degradation and genome instability in cells lacking BRCA1/2 [59,140], this genome maintenance pathway seems to be detrimental in *BRCA1/2*-deficient cells, suggesting an additional mechanism of synthetic lethality induced by PARPi. Remarkably, PARP1 depletion before BRCA1/2 loss restores stalled fork stability and even allows synthetic viability in mouse embryonic stem cells [60,140].

Multiple studies showed that restoring fork stabilization alone without restoring HR can drive the instauration of resistance to PARPi [59,60,61]. Interestingly, resistance to PARPi is associated with tolerance to replication stress-inducing chemotherapeutics such as cisplatin and topotecan [59]. Tobalina et al. [41] detected two reversion mutations in the isomerization region of BRCA1 that probably confer PARPi resistance through restoration of fork stabilization. Depletion of RADX, a RAD51 antagonist, restores stalled fork stability and induces chemoresistance in *BRCA2*-defective cells, probably due to enhanced association of RAD51 with stalled forks [61]. Inactivation of the remodelling protein SMARCAL1 confers chemoresistance, suppressing fork reversal and thus avoiding MRE11-dependent fork degradation [60]. Finally, clinical data revealed that low expression of RADX and SMARCAL1 correlates with poorer prognosis of *BRCA1/2*-mutated cancer patients, confirming the important role of fork stabilization in modulating chemosensitivities [59,60,61].

## 4. Future Perspectives

The discovery of multiple interactions between PARP1, DNA-damage repair pathways, and response to replication stress reinforces the increasing evidence that PARPi action is not limited to the inhibition of PARP1 catalytic activity. In this scenario, combination therapies could increase the sensitivity or overcome the resistance of certain tumours to PARPi, targeting pathways other than HR in which PARP1 is involved, both directly and indirectly. Considering that wild-type PTEN tumours could demonstrate relative resistance to PARPi and that the PI3K/AKT pathway is constitutively active in *BRCA1*-defective human cancer cells [73], the combination of PTEN-related PI3K/AKT pathway inhibitors with PARPi could be therapeutically beneficial. PI3K inhibitors in combination with PARPi have been shown to have synergistic therapeutic effects in *BRCA1*-deficient breast cancer in PDX models [141]. Targeting the replication checkpoint kinase ATR could represent another strategy. Combination of ATR inhibitors (ATRi) with PARPi in ATM-deficient *BRCA1/2*-mutated tumours achieved a therapeutic response at lower concentrations than monotherapy with either drug in vitro [138]. In addition, inhibition of ATR has the potential to reverse PARPi resistance provoked by reversions in *BRCA1/2*, as it limits HR acting on PALB2 [142]. More than a hundred ongoing clinical trials are investigating the combination of PARPi with immunotherapies and other agents, such as inhibitors of the replication stress, particularly ATRi [143] (Appendix A). On the other hand, clinical trials involving MDR1 inhibitors have provided poor results, possibly due to the important endogenous function of MDR1 in cell-mediated immune responses as recently revealed [144]. Since FANC2 is upregulated in *BRCA1/2*-mutated cancers, therapeutic targeting of the FA pathway might be required to potentiate the PARPi or platinum-based treatment for *BRCA1/2*-mutated tumours [14].

Several different mechanisms of resistance to PARPi have been proposed, although only reversion mutations in *BRCA1* and *BRCA2* have been proved in the clinical setting. More studies are needed to investigate the clinical impact of the other suggested mechanisms of resistance, up to now highlighted only in cell lines or PDX models. The implementation of liquid biopsy, in particular the analysis of the ctDNA at the diagnosis and during the treatment, could help to identify subclones of tumour cells primarily resistant to PARPi and to monitor the onset of secondary resistance before the clinical progression. The development of combination therapies probably will be necessary to overcome PARPi resistance and would hopefully identify additional settings for the use of PARPi.

## Figures and Tables

**Figure 1 cancers-14-02804-f001:**
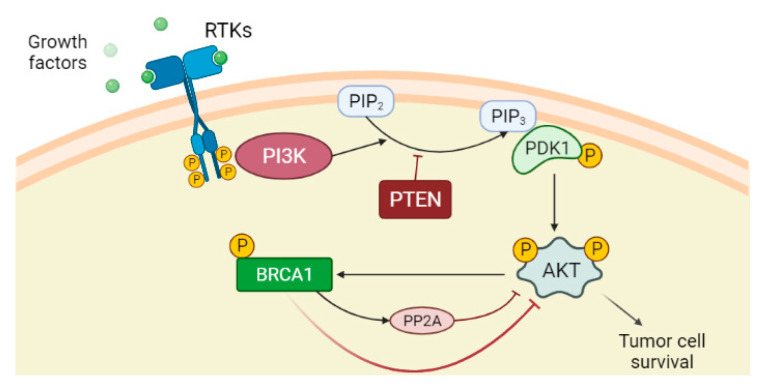
PI3K/AKT pathway is an intracellular signal transduction pathway that promotes cell growth and proliferation in response to extracellular signals. The binding of the ligands such as growth factors to the receptor tyrosine kinases (RTKs) induces dimerization of two RTK monomers, which consequently leads to activation of the intracellular tyrosine kinase domain and auto-phosphorylation by each monomer. The phosphatidylinositol 3-kinase (PI3K), once activated through direct stimulation of the regulatory subunit bound to the activated receptor, converts phosphatidylinositol 4,5-bisphosphate (PIP2) into phosphatidylinositol 3,4,5-triphosphate (PIP3). PIP3 binds the 3-phosphoinositide-dependent protein kinase-1 (PDK1) at the plasma membrane. PDK1 in turn phosphorylates and activates AKT protein. Once activated, AKT via phosphorylation regulates activation or suppression of several proteins involved in cell growth and proliferation. Phosphatase and tensin homolog (PTEN) is the main downregulation protein that can convert PIP3 into PIP2. Although AKT activation promotes BRCA1 expression through phosphorylation, BRCA1 can downregulate AKT activation by different mechanisms, among which are the activation of protein phosphatase 2A (PP2A), which dephosphorylates AKT.

**Figure 2 cancers-14-02804-f002:**
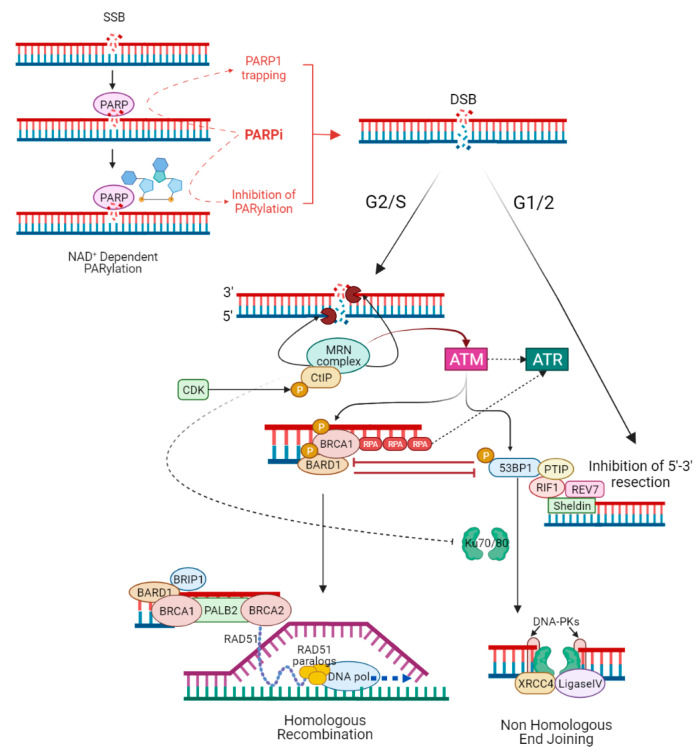
**PARP inhibitors and the interactions between homologous recombination and non-homologous end joining.** PARPi act mainly in a double way: inhibiting the catalytic activity of PARP1 (the so-called PARylation) and trapping PARP1 at sites of single-stranded DNA breaks (SSBs). In both cases, unrepaired SSBs lead to double-stranded DNA breaks (DSBs), which can be resolved mainly by HR or NHEJ. The choice between NHEJ and HR to repair DSBs is determined by several mechanisms, including activation of HR by cyclin-dependent kinase (CDK) activity (HR is restricted to G2/S phase when a homologous sister chromatid is available as template), or direct competition between HR and NHEJ stimulating factors at DSB sites. During G2/S, HR is activated by the binding of the MRN complex to DSB ends; MRN complex initiates DNA 5′-3′ end resection, leading to the formation of single-strand DNA (ssDNA) at the extremity of the DSB. CDK phosphorylated CtIP binds MRN complex to facilitate end resection. The ssDNA is protected from degradation by the loading of replication protein A (RPA). The MRN complex recruits and activates the protein kinase ATM, while RPA finally drives ATR activation. ATM phosphorylated BRCA1–BARD1 complex interacts with the bridging protein PALB2, which in turn promotes the recruitment of BRCA2. PALB2 and BRCA2 remove RPA and facilitate the assembly of the RAD51 nucleoprotein filament. RAD51 nucleoprotein filament mediates the invasion of ssDNA into the intact sister chromatid, searching for a homologous template for DNA synthesis and faithful repair of DNA. During G1/2, 5′-3′ end resection is suppressed and HR is inhibited due to lack of a sister chromatid. ATM phosphorylated 53BP1 binds and recruits RIF1 and PTIP that, together with the downstream effectors REV7 and Sheldin, inhibit 5′-3′ end resection and promote NHEJ. Ku70/80 heterodimer binds DSBs and recruits the DNA-dependent protein kinase catalytic subunit (DNA-PKcs) to form the DNA–PK complex. The latter engages XRCC4, XLF, and DNA ligase IV (LIG4) to align and ligates DNA ends regardless of sequence homology. BRCA1 antagonizes 53BP1 by stopping the translocation of RIF1 to DSBs in the G2/S phase, promoting HR; also, MRN/CtIP-dependent removal of Ku complex from DSBs favours HR.

**Figure 3 cancers-14-02804-f003:**
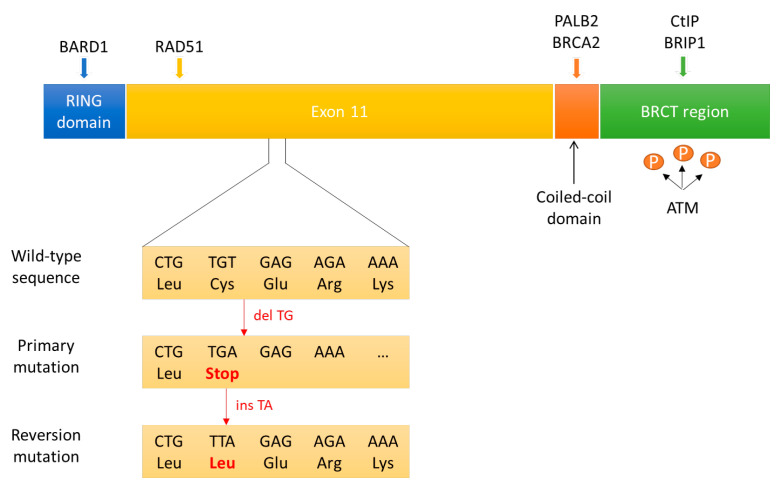
**BRCA1 structural domains and reversion mutations.** BRCA1 is a multi-domain protein. The N-terminal RING domain mediates interactions between BRCA1 and BARD1. Some N-terminal BRCA1 mutations may result in RING-less BRCA1 protein that could retain some residual activity in DNA damage response, conferring resistance to PARPi. The coiled-coil domain associates with PALB2 and BRCA2. The C-terminal domain (BRCT) contains potential ATM phosphorylation sites and can bind CtIP and BRIP1. C-terminal truncated BRCA1 proteins can be semifunctional, retaining the protein domains necessary to mediate interactions with PALB2-BRCA2-RAD51. Most amino acid sequences encoded by exon 11 are essential with regard to the protein function and for binding important HR proteins including RAD51. BRCA1 mRNA isoforms originated by alternative splicing and lacking exon 11 can produce truncated but hypomorphic proteins that have residual BRCA1 function. Primary mutations are mostly deletions causing frameshifts, leading to premature STOP codons. Reversion mutations in these cases are deletions or insertions (as in the example above), leading to the restoration of the BRCA open reading frame and functional protein.

**Figure 4 cancers-14-02804-f004:**
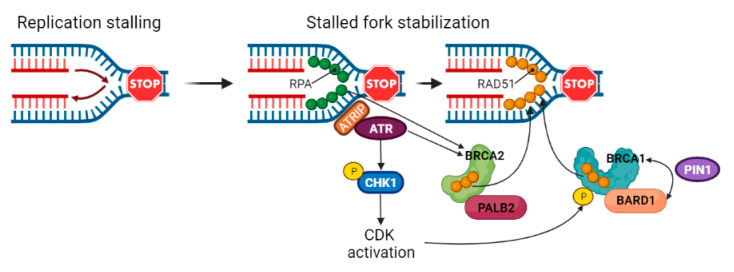
**Stalled fork stabilization.** During DNA duplication, replication forks often encounter obstacles that can lead to genotoxic fork stalling. Stalled forks are characterized by exposed DNA ends, which can be degraded by various cellular nucleases. Under replication stress, extensive ssDNA is readily protected and stabilised by RPA. ATRIP binding to RPA–ssDNA promotes ATR localization. ATR phosphorylates RPA and CHK1, leading to CDK activation. Both ATR and RPA promote PALB2 and BRCA2 translocation to stalled replication forks, while CDK phosphorylates BRCA1. The RPA–ssDNA complex is displaced by the RAD51 protein. RAD51 loading and stabilization are enhanced by the BRCA2/PALB2 complex as well as by the BRCA1–BARD1 complex. The phosphorylation-directed prolyl isomerase PIN1 enhances BRCA1–BARD1 interaction with RAD51. Stable RAD51 nucleoprotein filaments then mediate replication fork reversal.

**Table 1 cancers-14-02804-t001:** Proposed mechanisms of PARPi resistance.

Resistance Mechanism	Evidence	References
**Primary resistance**		
PI3K/AKT pathway activation	Cell lines	Yi et al. [31]
Wild-type PTEN	Cell lines	Dedes et al. [32]
Loss of NHEJ	Cell lines	Balmus et al. [33], Patel et al. [34], Mc Cormick et al. [35]
ALC1 overexpression	Cell lines	Juhász et al. [36]
**Secondary resistance**		
Upregulation of ABC transporters	Mouse models, cell lines	Jaspers et al. [37], Vaidyanathan et al. [38]
Decreased PARP1 trapping	Mouse models, cell lines	Pettitt et al. [39], Gogola et al. [40]
Restoration of HR		
-*BRCA1/2* reversion mutations	Tumour DNA and ctDNA from cancer patients	Tobalina et al. [41]
-Hypomorphic *BRCA1* allele	Cell lines, mouse models, PDXs	Drost et al. [42], Wang et al. [43], Cruz et al. [44], Wang et al. [45], Castroviejo-Bermejo et al. [46]
-Loss of *BRCA1* promoter methylation	Cell lines, PDXs	Ter Brugge et al. [47], Veeck et al. [48], Wang et al. [49]
-Loss of end resection regulation (53BP1, RIF1, REV7, Sheldin complex or DYNLL1 depletion)	Cell lines	Belotserkovskaya et al. [50], Xu et al. [51], Noordermeer et al. [52], Gupta et al. [53], He et al. [54]
-RAD51 overexpression	Ovarian cancer samples, cell lines	Kondrashova et al. [55], Liu et al. [56], Marzio et al. [57]
Stabilization of stalled fork (FANCD2 overexpression, RADX depletion, SMARCAL1 inactivation,)	Cell lines	Michl et al. [58], Chaudhuri et al. [59], Taglialatela et al. [60], Dungrawala et al. [61]

NHEJ: non-homologous end-joining. PDXs: patient-derived xenografts.

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
