# Peer review of "Insights into the Possible Molecular Mechanisms of Resistance to PARP Inhibitors"

_cancers, 2022, doi:10.3390/cancers14112804_

Round 1
Reviewer 1 Report
I actually do not have anything to comment on in this review. It is well written, easy to follow, includes clear explanations in text and figures. Overall, it is comprehensive and contributes a lot to the knowledge in this field.
Author Response
Many thanks, it is a great pleasure to receive such positive comment.
Reviewer 2 Report
Fig. 2 is similar to Fig. 1 of Cortesi, L., Piombino, C., & Toss, A. (2021). Germline Mutations in Other Homologous Recombination Repair-Related Genes Than BRCA1/2: Predictive or Prognostic Factors?. Journal of personalized medicine, 11(4), 245. https://doi.org/10.3390/jpm11040245.
Please cite the ref.
The authors mention cancers those have mutation of BRACA 1 and 2. How many percentages of cancers have the mutations?
Author Response
Many thanks for the positive comment.
I included the reference "Cortesi, L., Piombino, C., & Toss, A. (2021). Germline Mutations in Other Homologous Recombination Repair-Related Genes Than BRCA1/2: Predictive or Prognostic Factors?. Journal of personalized medicine, 11(4), 245."
In the introduction, data regarding prevalence of BRCA1/2 mutations in breast, ovarian, pancreatic, and prostate cancers have been added.
Reviewer 3 Report
In this manuscript entitled “Insights into the molecular mechanisms of resistance to PARP inhibitors” Piombino and Cortsesi provide a comprehensive review of the literature regarding the molecular mechanisms of primary and secondary resistance to PARP inhibitors.
The topic is very interesting and there are currently only few published studies related to resistance on PARP1 inhibitors. However, the manuscript is well structured and written and manages to provide a comprehensive review of the related mechanisms.
Minor comments:
1) Page 10, please correct the paragraph title (hypermethylation).
2) Future perspectives: A table with information related to clinical trials that combine PARP inhibitors with other agents would help to summarize the current efforts to battle resistance to PARP inhibitors
Author Response
Many thanks for the positive feedback. I corrected the title of paragraph at page 10. Since there are 111 ongoing clinical trials investigating the combination of PARPi with immunotherapies and other agents, I added a supplementary table (S1) with the first 100 results of the reasearch "PARP inhibitor AND combination" from clinicaltrials.gov